# Polymorphisms of *CYP7A1* and *HADHB* Genes and Their Effects on Milk Production Traits in Chinese Holstein Cows

**DOI:** 10.3390/ani14091276

**Published:** 2024-04-24

**Authors:** Ao Chen, Qianyu Yang, Wen Ye, Lingna Xu, Yuzhan Wang, Dongxiao Sun, Bo Han

**Affiliations:** 1Department of Animal Genetics and Breeding, College of Animal Science and Technology, Key Laboratory of Animal Genetics, National Engineering Laboratory for Animal Breeding, State Key Laboratory of Animal Biotech Breeding, China Agricultural University, Breeding and Reproduction of Ministry of Agriculture and Rural Affairs, Beijing 100193, China; capudrooks@foxmail.com (A.C.); 2021304010308@cau.edu.cn (Q.Y.); s20233040757@cau.edu.cn (W.Y.); xulingna@caas.cn (L.X.); yuzhanwang@cau.edu.cn (Y.W.); sundx@cau.edu.cn (D.S.); 2Beijing Jingwa Agricultural Innovation Center, Beijing 100193, China

**Keywords:** *CYP7A1*, *HADHB*, SNP, genetic effect, milk-production traits, dairy cattle

## Abstract

**Simple Summary:**

This study aimed to identify genetic variants related to differences in milk production in Chinese Holstein cows. By analyzing specific genes involved in lipid metabolism, researchers identified variants linked to the milk, fat, and protein yield cows produce over time. They also found patterns in these genetic variations that could affect how the genes work together. The results of this study suggest that these genetic markers could be useful for selecting cows with better milk-production traits. Additionally, the findings could help scientists better understand how these genes influence milk production, potentially leading to improved strategies for breeding dairy cattle. Overall, this research provides valuable insights into enhancing milk production in dairy farming, which could ultimately benefit both farmers and consumers.

**Abstract:**

Our preliminary research proposed the cytochrome P450 family 7 subfamily A member 1 (*CYP7A1*) and hydroxyacyl-coenzyme A dehydrogenase trifunctional multienzyme complex beta subunit (*HADHB*) genes as candidates for association with milk-production traits in dairy cattle because of their differential expression across different lactation stages in the liver tissues of Chinese Holstein cows and their potential roles in lipid metabolism. Hence, we identified single-nucleotide polymorphisms (SNPs) of the *CYP7A1* and *HADHB* genes and validated their genetic effects on milk-production traits in a Chinese Holstein population with the goal of providing valuable genetic markers for genomic selection (GS) in dairy cattle, This study identified five SNPs, 14:g.24676921A>G, 14:g.24676224G>A, 14:g.24675708G>T, 14:g.24665961C>T, and 14:g.24664026A>G, in the *CYP7A1* gene and three SNPs, 11:g.73256269T>C, 11:g.73256227A>C, and 11:g.73242290C>T, in *HADHB*. The single-SNP association analysis revealed significant associations (*p* value ≤ 0.0461) between the eight SNPs of *CYP7A1* and *HADHB* genes and 305-day milk, fat and protein yields. Additionally, using Haploview 4.2, we found that the five SNPs of *CYP7A1* formed two haplotype blocks and that the two SNPs of *HADHB* formed one haplotype block; notably, all three haplotype blocks were also significantly associated with milk, fat and protein yields (*p* value ≤ 0.0315). Further prediction of transcription factor binding sites (TFBSs) based on Jaspar software (version 2023) showed that the 14:g.24676921A>G, 14:g.24675708G>T, 11:g.73256269T>C, and 11:g.73256227A>C SNPs could alter the 5′ terminal TFBS of the *CYP7A1* and *HADHB* genes. The 14:g.24665961C>T SNP caused changes in the structural stability of the mRNA for the *CYP7A1* gene. These alterations have the potential to influence gene expression and, consequently, the phenotype associated with milk-production traits. In summary, we have confirmed the genetic effects of *CYP7A1* and *HADHB* genes on milk-production traits in dairy cattle and identified potential functional mutations that we suggest could be used for GS of dairy cattle and in-depth mechanistic studies of animals.

## 1. Introduction

Milk, often referred to as “white gold”, plays a pivotal role as a rich source of essential nutrients supporting growth, bone-mass accrual, and immune function, containing proteins, fats, carbohydrates, minerals, and vitamins [1]. The surge in demand for high-quality dairy products is attributed to factors like global population growth, increasing income levels, and shifting dietary preferences [2]. This change in demand is particularly pronounced in developing regions, notably in Asia and Africa. Enhancing yield and quality traits represent paramount objectives for dairy breeding, necessitating the implementation of effective breeding strategies such as genomic selection (GS) to optimize these characteristics.

GS facilitates early and precise selection of young bulls and reserve heifers without relying on phenotypic data, thereby shortening the dairy breeding cycle from 5–6 years to about 2 years. By significantly reducing generation intervals, this approach lowers breeding costs and accelerates genetic progress. Therefore, improved accuracy in GS is crucial for the development of improved breeding strategies [3]. Single-nucleotide polymorphisms (SNPs), which are prevalent in genomic regions like promoters or enhancers, serve as primary genetic markers. They impact gene expression by altering how regulatory proteins bind or how chromatin is structured, thus affecting protein structure, RNA splicing [4], and miRNA binding [5]. They are crucial in influencing pathways associated with milk synthesis and lactation [6,7,8,9]. Integrating functional SNPs into genomic-selection chips can enhance the efficiency, accuracy, and sustainability of selection [10], leading to improved milk-production traits in cows.

The rapid development of -omics technology has facilitated the process of functional-gene/locus mining. Our preliminary research proposed that the cytochrome P450 family 7 subfamily A member 1 (*CYP7A1*) and hydroxyacyl-coenzyme A dehydrogenase trifunctional multienzyme complex beta subunit (*HADHB*) genes are differentially expressed in different lactation stages in dairy cows and are involved in lipid metabolism [11]. The *CYP7A1* gene encodes cholesterol 7 alpha-hydroxylase, the rate-limiting enzyme in the classic pathway of bile-acid synthesis in the liver [12]. Bile acids are integral to processes such as lipid digestion, absorption, and excretion, as well as to the regulation of cholesterol homeostasis and metabolic pathways. The *HADHB* gene encodes hydroxyacyl-CoA dehydrogenase, a vital component of the mitochondrial trifunctional protein (MTP) complex, which is responsible for catalyzing the final three steps of mitochondrial beta-oxidation of long-chain fatty acids [13]. Furthermore, the *CYP7A1* gene is positioned within the quantitative trait loci (QTL) QTL_ID: 2732 and QTL_ID: 3408, which have been demonstrated to affect milkfat yield and percentage and milk protein percentage [14,15,16]. *HADHB* is situated within the QTL region related to milk yield (QTL_ID: 1512) and is approximately 11.8 kb from the SNP rs110711742, which is associated with milk yield [17]. These findings suggest a potential association between *CYP7A1* and *HADHB* and milk-production traits.

This study investigates the genetic associations between candidate genes *CYP7A1* and *HADHB* and milk-production traits in dairy cattle, including 305-day milk yield, fat yield, fat percentage, protein yield, and protein percentage. Additionally, predictive analyses were used to assess the regulatory effects of SNPs, particularly the effects on transcription factor binding sites (TFBSs) and mRNA stability, offering insights to support further exploration of causal mutations and their potential applications to developing GS chips for the breeding of dairy cattle.

## 2. Materials and Methods

### 2.1. Animal Selection and Phenotypic Data Collection

We selected 898 dairy cows from 45 Chinese Holstein sire families on 22 farms in Beijing Sunlon Livestock Development Co., Ltd. (Beijing, China) as the experimental population. The 45 bulls were used for SNP identification, and 898 cows were used for association analysis (898 cows in the first lactation and 611 in the second lactation). Each sire family had an average of 21 daughters, and each cow had three generations of pedigree information and Dairy Herd Improvement (DHI) records that were provided by the Beijing Dairy Cattle Centre (Beijing, China) (Appendix A). The cows in each sire family were distributed across various dairy farms and were maintained with the same feeding conditions. Data consisted of the milk-production phenotype of each cow for the whole lactation period of the parity, which comprised 305-day milk yield, fat yield, fat percentage, protein yield, and protein percentage. The study was conducted in accordance with Guide for the Care and Use of Laboratory Animals and was approved by the Institutional Animal Care and Use Committee (IACUC) at China Agricultural University (Beijing, China).

### 2.2. Genomic DNA Extraction

The frozen semen of 45 bulls and blood samples from 898 cows were provided by Beijing Dairy Cattle Center (Beijing, China). We extracted genomic DNA from the semen using the salting-out procedure and used a TIANamp Blood DNA Kit (Tiangen, Beijing, China) to extract DNA from the blood samples. The quantity and quality of extracted DNA were determined by a NanoDrop2000 spectrophotometer (Thermo Science, Hudson, NH, USA) and gel electrophoresis (1.5%), respectively.

### 2.3. Identification and Genotyping of SNPs in Candidate Genes

According to the genomic sequences of the two genes, *CYP7A1* (NC_037341.1) and *HADHB* (NC_037338.1) of *Bos taurus* in Genbank, primers were designed using Primer3.0 (https://bioinfo.ut.ee/primer3-0.4.0/ (accessed on 8 September 2021)). They were then synthesized by Beijing Liuhe Bgi Co., Ltd. (Beijing, China). The primers covered the entire coding region and 2000 base pairs upstream and downstream of the regulatory regions of the genes. The DNA samples from semen were diluted to 50 ng/µL and used for PCR amplification (Appendix A). PCR products were analyzed by 2% gel electrophoresis, and the qualified products were sequenced bidirectionally by Beijing Qingke Xinye Biotechnology Co., Ltd. (Beijing, China). Then, the sequences were aligned to the reference sequence (ARS-UCD1.2) using NCBI-blast (https://blast.ncbi.nlm.nih.gov/Blast.cgi (accessed on 20 November 2023)) to identify SNPs. Subsequently, we genotyped the SNPs in the 898 cows using Genotyping by Target Sequencing (GBTS) technology by Boruidi Biotechnology Co., Ltd. (Shijiazhuang, China). The allelic and genotypic frequencies were calculated, and the Hardy−Weinberg equilibrium was tested by the chi-squared test.

### 2.4. Linkage Disequilibrium (LD) Estimation

The extent of LD between the identified SNPs was estimated using Haploview 4.2 (Broad Institute of MIT and Harvard, Cambridge, MA, USA), with the solid spine algorithm. The haplotype block with a frequency greater than 0.05 was retained. The extent of LD is measured by the D′ value, to which it is proportional.

### 2.5. Association Analysis between SNP/Haplotype and Milk-Production Traits

The MIXED procedure in SAS 9.4 software (SAS Institute Inc., Cary, NC, USA) was used to perform association analysis between the SNP or haplotype block and the five milk-production traits (305-day milk yield, fat yield, fat percentage, protein yield and protein percentage) using the following animal model:y=μ+HYS+b×M+G+a+e
where y is the phenotypic value of each trait of each cow; μ is the overall mean; HYS is the fixed effect of farm (1–22: 22 farms), calving year (1–4: 2012–2015) and calving season (1: April–May; 2: June–August; 3: September–November and 4: December–March); M is the age at calving as a covariant, b is the regression coefficient of covariant M; G is the genotype or haplotype combination effect; a is the individual random additive genetic effect with a distribution of $N\left(0,I\delta_{e}^{2}\right)$; A is a pedigree-based relationship matrix with an additive genetic variance of $\delta_{a}^{2}$; and e is the random residual with a distribution of $N\left(0,I\delta_{e}^{2}\right)$, where I is the unit matrix and $\delta_{e}^{2}$ is the residual variance. A Bonferroni correction was carried out by multiple tests with a significance level equal to the original *p* value divided by the number of genotype or haplotype combinations.

Furthermore, the additive, dominant, and allelic substitution effects of the SNPs were calculated by the following formulas:$$a=\left(X_{AA}-X_{BB}\right)/2;$$
(1)$$d=X_{AB}-\left(X_{AA}+X_{BB}\right)/2;$$
(2)$$\alpha =a+d\left(q-p\right),$$
where a is the additive effect; d is the dominant effect; α is the allelic substitution effect; $X_{AA}$, $X_{AB}$ and $X_{BB}$ are the least-squares means of milk-production traits for the corresponding genotypes; p is the frequency of allele A; and q is the frequency of allele B.

### 2.6. Prediction of Transcription Factor Binding Sites

We used Jaspar (http://jaspar.genereg.net/ (version 2023)) software to predict whether SNPs in the 5′ flanking region of *CYP7A1* and *HADHB* genes changed the transcription factor binding sites (TFBS) (relative score ≥ 0.80).

### 2.7. Prediction of mRNA Structure

We used the NCBI database (https://blast.ncbi.nlm.nih.gov/ (accessed on 20 November 2023)) to query for synonymy or missense mutations on exons. Subsequently, we used RNAfold WebServer (http://rna.tbi.univie.ac.at/cgi-bin/RNAWebSuite/RNAfold.cgi (accessed on 20 November 2023)) to predict the changes to the RNA secondary structure caused by the mutation in the untranslated region, with the parameters of minimum free energy (MFE), partition function, and avoidance of isolated base pairs. MFE was used for a direct comparison of the folding stability of RNAs of the same sizes; the smaller the MFE, the greater the stability.

## 3. Results

### 3.1. SNPs Identification

We identified five SNPs in the *CYP7A1* gene and three in the *HADHB* gene in 898 dairy cows from 45 Chinese Holstein sire families on 22 farms. In *CYP7A1*, 14:g.24676921A>G (rs42765357), 14:g.24676224G>A (rs109454495) and 14:g.24675708G>T (rs42765359) were located in the 5′ flanking region; 14:g.24665961C>T (rs108958186), was found in the 3′ untranslated region (UTR); and 14:g.24664026A>G (rs109680813) was found in the 3′ regulatory region. In *HADHB*, 11:g.73256269T>C (rs110033443) was located in the 5′ flanking region, 11:g.73256227A>C (rs134856746) in the 5′ UTR, and 11:g.73242290C>T (rs137211407) in the intron between exons 5 and 6. The genotypic and allelic frequencies of all the identified SNPs are summarized in Table 1.

### 3.2. Single-Marker Association Analysis

The results of the association analysis (Table 2) showed that 14:g.24665961C>T, in *CYP7A1*, was significantly associated with milk yield (*p* value = 0.0443) and fat yield (*p* value ≤ 0.0001) in the first lactation period; four SNPs, 14:g.24676921A>G, 14:g.24676224G>A, 14:g.24675708G>T, and 14:g.24665961C>T, were significantly associated with milk, fat and protein yields (*p* value ≤ 0.0059) and 14:g.24664026A>G was highly significantly associated with fat yield (*p* value = 0.0037) in the second lactation.

In *HADHB*, all three SNPs, 11:g.73256269T>C, 11:g.73256227A>C, and 11:g.73242290C>T, were significantly associated with milk and protein yields in the first lactation (*p* value ≤ 0.0461) and with milk, fat, and protein yields in the second lactation (*p* value ≤ 0.001). Additionally, the results of the additive, dominant, and allelic substitution effects were shown in Table 3.

### 3.3. Haplotype Association Analysis

Using Haploview 4.2, five SNPs in *CYP7A1*, 14:g.24675708G>T, 14:g.24676224G>A, 14:g.24676921A>G, 14:g.24665961C>T and 14:g.24664026A>G, formed two haplotype blocks (Figure 1). In Block 1, the frequencies of haplotypes H1 (AC), H2 (GT) and H3 (AT) were 0.463, 0.342, and 0.194, respectively. Block 2 was composed of three haplotypes, H1 (TGA), H2 (GAG), and H3 (TGG), with frequencies of 0.349, 0.545, and 0.105, respectively. We found that the two blocks in *CYP7A1* were significantly associated with milk, fat, and protein yields in both lactations (*p* value ≤ 0.0011; Table 4).

Similarly, two SNPs, 11:g.73256269T>C and 11:g.73256227A>C, in *HADHB* formed Block 3 (Figure 1). The frequencies of haplotypes H1 (AT), H2 (CC), and H3 (CT) were 0.159, 0.806, and 0.035, respectively. This block was found to be associated with milk, fat, and protein yields in the first lactation (*p* value ≤ 0.0315), and milk, fat, and protein yields in the second lactation (*p* value ≤ 0.0052; Table 4).

### 3.4. Changes of Transcription Factor Binding Sites Caused by SNPs in 5′ Region

Using the JASPAR, changes in TFBS were predicted for four SNPs in the 5′ regulatory region of the *CYP7A1* and *HADHB* genes. As a result, it was found that, in *CYP7A1*, the allele G of 14:g.24676921A>G created a binding site (BS) for transcription factors (TFs) EBF1 and SP1, while the mutation from G to T of 14:g.24675708G>T eliminated the BS for ELK1 but created a BS for FOXC1 and ZNF354C. For *HADHB*, before and after mutation, 11:g.73256269T>C resulted in the disappearance of the BS for PAX2 and NR2F1 and the appearance of that for TFAP2A; in addition, the allele A of 11:g.73256227A>C created a BS for ZNF354C and allele C created a BS for HINFP (Table 5).

### 3.5. mRNA Structural Stability Altered by the Mutation in Untranslated Region

Using RNAfold, changes in the mRNA minimum free energy (MFE) were predicted. The mutation of C to T in SNP 14:g.24665961C>T causes the MFE of mRNA to change from −578.10 kcal/mol to −539.20 kcal/mol, resulting in a decrease in the structural stability of *CYP7A1* mRNA.

## 4. Discussion

The *CYP7A1* and *HADHB* genes have been implicated in milk-production traits in dairy cows, as indicated by recent research studies. *CYP7A1* is known to influence triglyceride and cholesterol metabolism in the liver tissue of dairy cows and is a rate-limiting enzyme for bile-acid synthesis [18]; in chickens, *HADHB* plays a crucial role in liver lipid metabolism, particularly in inducing peroxisomal and mitochondrial β-oxidation activity [19]. These findings affirm the possible beneficial influence of these genes on milk production. Here, further identification of SNPs in the *CYP7A1* and *HADHB* genes and analysis of their association with milk-production traits in dairy cows provide valuable information for understanding the mechanism of inheritance of these traits and linking the metabolic functions of these genes to their phenotypic effects. Furthermore, the integration of significant genetic sites from the *CYP7A1* and *HADHB* genes into GS models can facilitate more precise selection of individuals with favorable milk-production traits. The purpose of these models is to predict the genetic advantage of the trait of interest based on the SNP profile of the individual, such that breeders can screen individuals for specific genetic variants associated with particular milk-production phenotypes, select them for breeding, and thus improve the efficiency of breeding.

In this study, we identified SNPs in the *CYP7A1* and *HADHB* genes and confirmed their significant associations with milk-production traits in dairy cows. SNPs can cause phenotypic variation by affecting the function or expression of genes involved in various biological processes [20,21,22,23]. This study did find significant differences in the milk-production phenotypes of cows with different SNP sites in the *CYP7A1* and *HADHB* genes. The number of individual cows with lower phenotypic values was also smaller, possibly because individuals of genotypes with higher-production phenotypes had been selected for while those with lower-production phenotypes had been eliminated during artificial breeding over the long term [24]. In addition, the phenotypic values of the cows and the significance of SNP/haplotype block associations were lower in the first lactation than in the second lactation. This difference could be explained by either of two possible causes. First, the varying numbers of cows in the two lactation periods may impact the statistical significance of the corresponding results. Second, there are physiological differences between the two lactation periods, as dairy cows tend to produce more milk in their second lactation [25].

Transcription factors are proteins that bind to specific DNA sequences and modulate the expression of target genes involved in various biological processes [26]. In this study, we found changes in TFBSs caused by SNPs in the 5′ flanking regions of the *CYP7A1* and *HADHB* genes. These changes may result in altered regulation of gene expression by TFs, affecting the participation of the corresponding expression products in normal physiological and metabolic activities and thus, in turn, contributes to changes in milk production. For instance, in *HADHB*, TFAP2A is a specific nuclear transcription factor that may promote cell proliferation by positively regulating certain cell-proliferation-associated proteins such as EIA, SV40, and c-myc [27]. This TF binding with 11:g.73256269C may promote expression of the *CYP7A1* gene and explain the significantly higher milk, fat and protein yields in the cows with genotype CC compared to those with genotype TT in the second lactation. Conversely, ZNF354C binds to form a number of transcriptional-repression complexes that inhibit the transcription of the intended target genes [28]. Binding of the *HDAHB* gene to this TF at 11:g.73256227A may result in the repression of its expression, which may have contributed to the significantly lower production of milk, fat, and milk proteins by the individuals with genotype AA than by those with CC in the second lactation.

Although the region is situated within gene exons, the term “untranslated region” (UTR) refers to the non-coding segment located at the mRNA molecule’s terminus. Consequently, SNPs found within the UTR do not exert a direct influence on transcription and translation processes and are typically regarded as neutral variants. However, despite their lack of direct impact on encoded proteins, such SNPs in UTRs may change the mRNA’s minimum free energy, leading to reduced mRNA stability. The synonymous mutation 14:g.24665961C>T, located in the untranslated region of the *CYP7A1* gene, reduces mRNA stability, resulting in reduced efficiency of transcription and translation of this gene, possibly affecting triglyceride and cholesterol metabolism [29,30], which may explain why the amount of milkfat yield in second lactation was significantly higher in the individuals with the genotype CC than in those with the genotype TT at the SNP site 14:g.24665961C>T.

## 5. Conclusions

This study identified polymorphisms in the *CYP7A1* and *HADHB* genes and their significant genetic effects on milk-production traits in Chinese Holstein cows, offering genetic markers for GS of dairy cattle; thus, four SNPs that altered TFBSs, 14:g.24676921A>G, 14:g.24675708G>T, 11:g.73256269T>C, and 11:g.73256227A>C, were proposed as possible causative mutations for the formation of milk-production traits. Further in-depth verification is needed before these SNPs can be used to provide genetic information to support the breeding of dairy cattle.

## Figures and Tables

**Figure 1 animals-14-01276-f001:**
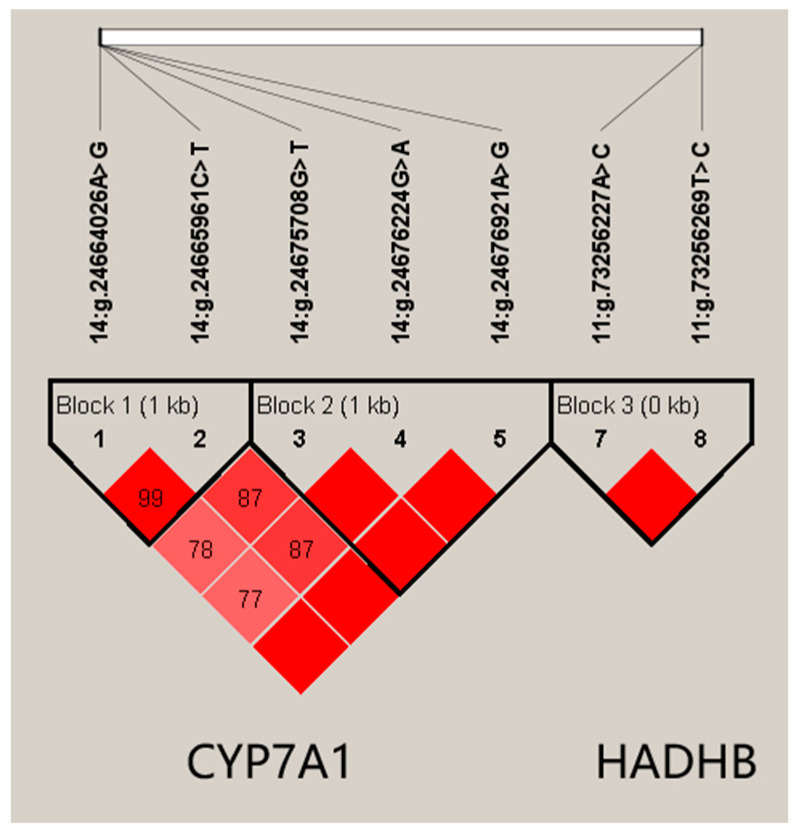
Linkage disequilibrium estimated between SNPs in *CYP7A1* and *HADHB* gene. The blocks indicate haplotype blocks, and the text above the horizontal numbers is the SNP names. The values in the red boxes are pair-wise SNP correlations (D′, while bright red boxes without numbers indicate complete linkage disequilibrium (LD) (D′ = 1)).

**Table 1 animals-14-01276-t001:** Detailed information about the identified SNPs and their genotypic and allelic frequencies.

Gene	SNP	rs ID	Gene Region	Genotype	Genotypic Frequency	Allele	Allelic Frequency
*CYP7A1*	14:g.24676921A>G	rs42765357	5′ flanking region	AA	0.1341	A	0.3511
				AG	0.4340	G	0.6489
				GG	0.4319		
	14:g.24676224G>A	rs109454495	5′ flanking region	AA	0.3073	A	0.5433
				AG	0.4720	G	0.4567
				GG	0.2207		
	14:g.24675708G>T	rs42765359	5′ flanking region	GG	0.3083	G	0.5444
				GT	0.4720	T	0.4566
				TT	0.2196		
	14:g.24665961C>T	rs108958186	3′ UTR	CC	0.2218	C	0.4683
				CT	0.4931	T	0.5317
				TT	0.2851		
	14:g.24664026A>G	rs109680813	3′ flanking region	AA	0.4298	A	0.6584
				AG	0.4572	G	0.3416
				GG	0.1130		
*HADHB*	11:g.73256269T>C	rs110033443	5′ flanking region	CC	0.6410	C	0.8031
				CT	0.3242	T	0.1969
				TT	0.0348		
	11:g.73256227A>C	rs134856746	5′ UTR	AA	0.0201	A	0.1605
				AC	0.2809	C	0.8395
				CC	0.6990		
	11:g.73242290C>T	rs137211407	Intron	CC	0.1373	C	0.2450
				CT	0.2154	T	0.7550
				TT	0.6473		

UTR: untranslated region.

**Table 2 animals-14-01276-t002:** Associations of eight SNPs in the *CYP7A1* and *HADHB* genes with milk yield and milk-composition traits in Chinese Holstein cattle during the first and second lactations.

Gene	SNP	Lactation	Genotype (No.)	Milk Yield (kg)	Fat Yield (kg)	Fat Percentage (%)	Protein Yield (kg)	Protein Percentage (%)
*CYP7A1*	14:g.24676921A>G	1	AA (118)	10,289 ± 64.59	345.06 ± 2.27	3.37 ± 0.03	306.85 ± 2.73	2.98 ± 0.01
			AG (391)	10,308 ± 43.99	344.94 ± 1.55	3.36 ± 0.03	308.93 ± 2.17	2.30 ± 0.01
			GG (389)	10,299 ± 43.98	345.75 ± 1.55	3.37 ± 0.03	308.37 ± 2.18	3.00 ± 0.01
			*p* value	0.9512	0.8719	0.8957	0.6648	0.3686
		2	AA (79)	11,067 ^A^ ± 103.8	396.65 ^A^ ± 4.38	3.59 ± 0.04	330.55 ^Aa^ ± 4.38	2.99 ± 0.01
			AG (272)	10,660 ^B^ ± 69.92	378.52 ^B^ ± 3.10	3.56 ± 0.03	317.36 ^Bb^ ± 3.10	2.98 ± 0.01
			GG (260)	10,717 ^B^ ± 71.91	383.4 ^B^ ± 3.17	3.59 ± 0.03	319.70 ^ABb^ ± 3.17	2.99 ± 0.01
			*p* value	0.0003	<0.0001	0.5719	0.0059	0.5828
	14:g.24676224G>A	1	AA (278)	10,287 ± 50.05	346.51 ± 1.73	3.38 ± 0.03	308.51 ± 2.31	3.01 ± 0.01
			GA (422)	10,277 ± 43.34	343.81 ± 1.49	3.36 ± 0.03	307.49 ± 2.15	2.99 ± 0.01
			GG (198)	10,370 ± 53.54	346.97 ± 1.85	3.37 ± 0.03	310.12 ± 2.41	3.00 ± 0.01
			*p* value	0.1972	0.1421	0.574	0.3774	0.2884
		2	AA (181)	10,755 ^B^ ± 78.19	385.15 ^AB^ ± 3.38	3.59 ± 0.03	321.69 ^AB^ ± 3.38	3.00 ± 0.01
			GA (293)	10,622 ^B^ ± 69.32	378.07 ^B^ ± 3.05	3.58 ± 0.03	315.65 ^B^ ± 3.05	2.97 ± 0.01
			GG (137)	10,942 ^A^ ± 84.75	389.83 ^A^ ± 3.63	3.57 ± 0.03	326.73 ^A^ ± 3.63	2.98 ± 0.01
			*p* value	0.0006	0.0011	0.8325	0.0029	0.0677
	14:g.24675708G>T	1	GG (279)	10,271 ± 50.23	346.01 ± 1.73	3.38 ± 0.03	308.12 ± 2.31	3.01 ± 0.01
			GT (422)	10,282 ± 43.57	343.92 ± 1.49	3.36 ± 0.03	307.55 ± 2.15	2.99 ± 0.01
			TT (197)	10,378 ± 53.8	347.33 ± 1.86	3.37 ± 0.03	310.48 ± 2.41	3.00 ± 0.01
			*p* value	0.1313	0.1635	0.5444	0.2926	0.244
		2	GG (181)	10,755 ^AB^ ± 78.19	385.15 ^AB^ ± 3.38	3.59 ± 0.03	321.69 ^AB^ ± 3.38	3.00 ± 0.01
			GT (293)	10,622 ^B^ ± 69.32	378.07 ^B^ ± 3.05	3.58 ± 0.03	315.65 ^B^ ± 3.05	2.97 ± 0.01
			TT (137)	10,942 ^A^ ± 84.76	389.83 ^A^ ± 3.63	3.57 ± 0.03	326.73 ^A^ ± 3.63	2.98 ± 0.01
			*p* value	0.0006	0.0011	0.8325	0.0029	0.0677
	14:g.24665961C>T	1	CC (198)	10,355 ± 53.37	348.72 ^A^ ± 1.87	3.39 ± 0.03	310.35 ^A^ ± 2.42	3.00 ± 0.01
			CT (438)	10,250 ± 42.29	341.82 ^B^ ± 1.48	3.35 ± 0.03	306.39 ^B^ ± 2.14	2.99 ± 0.01
			TT (262)	10,354 ± 50.21	348.93 ^A^ ± 1.76	3.38 ± 0.03	310.43 ^A^ ± 2.33	3.00 ± 0.01
			*p* value	0.0443	<0.0001	0.2811	0.0258	0.369
		2	CC (136)	10,975 ^Aa^ ± 85.38	394.69 ^Aa^ ± 3.66	3.60 ± 0.03	328.62 ^Aa^ ± 3.66	2.99 ± 0.01
			CT (305)	10,604 ^Bb^ ± 67.5	377.09 ^Bc^ ± 2.99	3.57 ± 0.03	315.25 ^Bb^ ± 2.99	2.98 ± 0.01
			TT (170)	10,813 ^ABa^ ± 80.72	385.16 ^ABb^ ± 3.49	3.57 ± 0.03	322.85 ^ABa^ ± 3.49	2.99 ± 0.01
			*p* value	<0.0001	<0.0001	0.5643	0.0001	0.1667
	14:g.24664026A>G	1	AA (385)	10,251 ± 42.60	344.86 ± 1.48	3.38 ± 0.03	306.82 ± 2.16	3.00 ± 0.01
			AG (410)	10,358 ± 42.67	346.02 ± 1.49	3.36 ± 0.03	309.88 ± 2.17	2.99 ± 0.01
			GG (103)	10,284 ± 68.88	344.17 ± 2.41	3.36 ± 0.04	309.52 ± 2.90	3.01 ± 0.01
			*p* value	0.0583	0.6425	0.5414	0.1429	0.5095
		2	AA (261)	10,798 ± 70.29	386.32 ^Aa^ ± 3.10	3.59 ± 0.03	322.31 ± 3.10	2.99 ± 0.01
			AG (286)	10,676 ± 69.67	378.68 ^Ab^ ± 3.09	3.57 ± 0.03	317.79 ± 3.09	2.98 ± 0.01
			GG (64)	10,758 ± 113.28	389.85 ^Aa^ ± 4.75	3.62 ± 0.05	320.94 ± 4.75	2.99 ± 0.02
			*p* value	0.2012	0.0037	0.4078	0.2597	0.7069
*HADHB*	11:g.73256269T>C	1	CC (580)	10,345 ± 40.5	345.52 ± 1.40	3.36 ± 0.03	309.82 ± 2.08	3.00 ± 0.01
			TC (287)	10,249 ± 49.36	346.32 ± 1.72	3.39 ± 0.03	306.67 ± 2.30	3.00 ± 0.01
			TT (31)	10,100 ± 118.79	338.41 ± 4.16	3.35 ± 0.06	302.28 ± 4.45	2.99 ± 0.02
			*p* value	0.0236	0.1926	0.2617	0.0461	0.8674
		2	CC (370)	10,873 ^A^ ± 66.55	387.93 ^Aa^ ± 2.95	3.58 ± 0.03	324.74 ^A^ ± 2.95	2.99 ± 0.01
			TC (214)	10,626 ^B^ ± 74.67	380.21 ^Ab^ ± 3.26	3.60 ± 0.03	315.93 ^B^ ± 3.26	2.98 ± 0.01
			TT (27)	10,091 ^C^ ± 162.53	351.42 ^Bc^ ± 6.68	3.50 ± 0.07	299.92 ^B^ ± 6.68	2.98 ± 0.02
			*p* value	<0.0001	<0.0001	0.3795	<0.0001	0.585
	11:g.73256227A>C	1	AA (18)	10,532 ^AB^ ± 156.37	349.98 ± 5.49	3.31 ± 0.08	313.59 ^b^ ± 5.73	3.00 ± 0.03
			AC (250)	10,178 ^B^ ± 51.16	344.16 ± 1.81	3.40 ± 0.03	304.93 ^a^ ± 2.36	3.00 ± 0.01
			CC (630)	10,333 ^A^ ± 39.75	345.45 ± 1.42	3.36 ± 0.03	309.37 ^b^ ± 2.07	3.00 ± 0.01
			*p* value	0.0015	0.499	0.2506	0.0184	0.9941
		2	AA (19)	9958.5 ^Ac^ ± 193.86	335.25 ^A^ ± 7.93	3.42 ± 0.08	294.18 ^B^ ± 7.93	2.98 ± 0.03
			AC (184)	10,540 ^Bb^ ± 78.1	377.62 ^B^ ± 3.39	3.60 ± 0.03	312.87 ^B^ ± 3.39	2.97 ± 0.01
			CC (408)	10,883 ^Aa^ ± 65.13	388.71 ^C^ ± 2.90	3.58 ± 0.03	325.20 ^A^ ± 2.90	2.99 ± 0.01
			*p* value	<0.0001	<0.0001	0.0665	<0.0001	0.404
	11:g.73242290C>T	1	CC (123)	10,254 ^AB^ ± 65.02	344.89 ± 2.30	3.38 ± 0.04	306.77 ^ab^ ± 2.73	2.99 ± 0.01
			CT (189)	10,168 ^B^ ± 54.74	344.54 ± 1.95	3.40 ± 0.03	304.97 ^b^ ± 2.47	3.00 ± 0.01
			TT (586)	10,358 ^A^ ± 40.35	345.66 ± 1.45	3.36 ± 0.03	309.91 ^a^ ± 2.08	3.00 ± 0.01
			*p* value	0.0012	0.8134	0.2182	0.0206	0.7748
		2	CC (88)	10,476 ^B^ ± 99.09	368.81 ^Bc^ ± 4.20	3.54 ± 0.04	308.42 ^B^ ± 4.20	2.96 ± 0.01
			CT (147)	10,545 ^B^ ± 82.76	379.36 ^ABb^ ± 3.58	3.62 ± 0.03	315.16 ^B^ ± 3.58	2.99 ± 0.01
			TT (376)	10,884 ^A^ ± 66.08	387.96 ^Aa^ ± 2.95	3.57 ± 0.03	325.03 ^A^ ± 2.95	2.99 ± 0.01
			*p* value	<0.0001	<0.0001	0.1681	<0.0001	0.0674

The number in the table represents the mean ± standard deviation; the number in the bracket represents the number of cows with the corresponding genotype; the *p* value shows the significance calculated for the genetic effects of SNPs; the superscript letters indicate the significance calculated for the effects different genotypes via the comparison of each pair of pairs; ^a^, ^b^, ^c^ within the same column with different superscripts means that the *p* value is < 0.05; and ^A^, ^B^, ^C^ within the same column with different superscripts means that the *p* value is <0.01.

**Table 3 animals-14-01276-t003:** Additive, dominant and allele substitution effects of 8 SNPs in the *CYP7A1* and *HADHB* genes on milk yield and milk-composition traits in Chinese Holstein cattle during two lactations.

Gene	SNP Name	Lactation	Effect	Milk Yield (kg)	Fat Yield (kg)	Fat Percentage (%)	Protein Yield (kg)	Protein Percentage (%)
*CYP7A1*	14:g.24676921A>G	1	Additive Effect (a)	−5.0162	−0.3432	−0.00126	−0.7593	−0.00757
			Dominance Effect (d)	14.2908	−0.467	−0.00879	1.3145	0.007425
			Allele Substitution Effect (α)	−9.3289	−0.2023	0.001392	−1.156	−0.00981
		2	Additive Effect (a)	174.61 **	6.6291 **	0.00234	5.4211 *	−0.00187
			Dominance Effect (d)	−232.24 **	−11.5028 **	−0.02803	−7.7643 **	−0.00847
			Allele Substitution Effect (α)	243.41 **	10.0366 **	0.01064	7.7212 **	0.000633
	14:g.24676224G>A	1	Additive Effect (a)	−41.8497	−0.2291	0.006524	−0.8039	0.004882
			Dominance Effect (d)	−51.9072	−2.9335 *	−0.018	−1.8257	−0.00847
			Allele Substitution Effect (α)	−46.4739	−0.4904	0.00492	−0.9666	0.004128
		2	Additive Effect (a)	−93.7702 *	−2.34	0.009039	−2.5205	0.007237
			Dominance Effect (d)	−226.82 **	−9.4213 **	−0.00727	−8.5594 **	−0.01769
			Allele Substitution Effect (α)	−110.1 *	−3.0184	0.008515	−3.1369	0.005963
	14:g.24675708G>T	1	Additive Effect (a)	−53.7569	−0.6575	0.0065	−1.1824	0.004344
			Dominance Effect (d)	−42.5283	−2.7457	−0.01911	−1.7491	−0.00997
			Allele Substitution Effect (α)	−57.6403	−0.9082	0.004755	−1.3421	0.003434
		2	Additive Effect (a)	−93.7702 *	−2.34	0.009039	−2.5205	0.007237
			Dominance Effect (d)	−226.82 **	−9.4213 **	−0.00727	−8.5594 **	−0.01769
			Allele Substitution Effect (α)	−110.1 *	−3.0184	0.008515	−3.1369	0.005963
	14:g.24665961C>T	1	Additive Effect (a)	0.911	−0.106	0.002497	−0.04286	−0.00055
			Dominance Effect (d)	−104.99 *	−6.9981	−0.03212	−3.9993 *	−0.01024
			Allele Substitution Effect (α)	8.3937	0.3928	0.004787	0.2422	0.000182
		2	Additive Effect (a)	80.7445	4.764 *	0.01721	2.8865	0.001474
			Dominance Effect (d)	−289.97 **	−12.8359 **	−0.01506	−10.4784 **	−0.0176
			Allele Substitution Effect (α)	96.8803 *	5.4783 **	0.01805	3.4696	0.002453
	14:g.24664026A>G	1	Additive Effect (a)	−16.3957	0.3409	0.009981	−1.3534	−0.00523
			Dominance Effect (d)	89.7199	1.5028	−0.0135	1.7121	−0.00847
			Allele Substitution Effect (α)	11.7791	0.8128	0.005741	−0.8157	−0.00789
		2	Additive Effect (a)	20.2446	−1.7674	−0.01594	0.6858	−0.00119
			Dominance Effect (d)	−101.99	−9.407 **	−0.03741	−3.8298	−0.00827
			Allele Substitution Effect (α)	−14.8297	−5.0024	−0.0288	−0.6313	−0.00403
*HADHB*	11:g.73256269T>C	1	Additive Effect (a)	122.03 *	3.5596	0.003985	3.7702	0.004206
			Dominance Effect (d)	26.4066	4.3527	0.03899	0.6195	0.001006
			Allele Substitution Effect (α)	104.04 **	0.5932	−0.02259	3.348 *	0.003521
		2	Additive Effect (a)	390.89 **	18.2568 **	0.03657	12.4128 **	0.001967
			Dominance Effect (d)	143.14	10.5408 **	0.05419	3.5935	−0.00841
			Allele Substitution Effect (α)	471.25 **	24.1742 **	0.067	14.4302 **	−0.00275
	11:g.73256227A>C	1	Additive Effect (a)	99.5398	2.2649	−0.02704	2.1125	−0.00149
			Dominance Effect (d)	−254.17 **	−3.5579	0.05961 *	−6.5477 *	0.001585
			Allele Substitution Effect (α)	272.76 *	4.6897	−0.06766	6.5749	−0.00257
		2	Additive Effect (a)	−462.42 **	−26.7305 **	−0.08246 *	−15.5101 **	−0.0063 **
			Dominance Effect (d)	118.71	15.6455 **	0.1017 *	3.184	−0.00734
			Allele Substitution Effect (α)	−538 **	−36.6914 **	−0.1472 *	−17.5372 **	−0.00162 **
	11:g.73242290C>T	1	Additive Effect (a)	−51.7106	−0.383	0.01289	−1.5714	−0.00237
			Dominance Effect (d)	−137.38	−0.7373	0.02938	−3.3741	0.0067
			Allele Substitution Effect (α)	19.1215	−0.00287	−0.00226	0.1682	−0.00583
		2	Additive Effect (a)	−203.9 **	−9.573 **	−0.01824	−8.3062 **	−0.01506 *
			Dominance Effect (d)	−135.25	0.9699	0.06063	−1.5693	0.01771
			Allele Substitution Effect (α)	−140.15 *	−10.0302 **	−0.04682	−7.5665 **	−0.02341 *

Note: The number in the table represents the mean; * indicates *p* < 0.05; ** indicates *p* < 0.01.

**Table 4 animals-14-01276-t004:** Associations of haplotype blocks in the *CYP7A1* and *HADHB* genes with milk yield and milk-composition traits in Chinese Holstein cattle during first and second lactations.

Block	Lactation	Haplotype Combination	Milk Yield (kg)	Fat Yield (kg)	Fat Percentage (%)	Protein Yield (kg)	Protein Percentage (%)
BLOCK1	1	H1H1 (197)	10,410 ^A^ ± 53.81	352.81 ^ABa^ ± 1.9	3.41 ± 0.03	312.82 ^A^ ± 2.41	3.01 ± 0.01
		H1H2 (282)	10,329 ^A^ ± 46.52	346.8 ^Bb^ ± 1.64	3.38 ± 0.03	310.03 ^AB^ ± 2.23	3.01 ± 0.01
		H1H3 (155)	10,125 ^B^ ± 57.38	343.65 ^Bb^ ± 2.02	3.41 ± 0.03	304.38 ^B^ ± 2.52	3.01 ± 0.01
		H2H2 (102)	10,314 ^AB^ ± 68.66	351.65 ^ABab^ ± 2.42	3.41 ± 0.04	310.75 ^AB^ ± 2.85	3.02 ± 0.01
		H2H3 (127)	10,466 ^A^ ± 63.22	356.37 ^Aa^ ± 2.23	3.42 ± 0.04	313.93 ^A^ ± 2.68	3.00 ± 0.01
		*p* value	<0.0001	<0.0001	0.6801	0.0011	0.7621
	2	H1H1 (136)	10,937 ^Aa^ ± 86.81	392.63 ^Aa^ ± 3.73	3.60 ± 0.03	327.99 ^A^ ± 2.71	3.00 ± 0.01
		H1H2 (201)	10,539 ^Bb^ ± 76.03	374.21 ^Bb^ ± 3.31	3.57 ± 0.03	316.09 ^B^ ± 2.40	3.00 ± 0.01
		H1H3 (104)	10,453 ^Bb^ ± 91.26	372.18 ^Bb^ ± 3.89	3.57 ± 0.04	311.78 ^B^ ± 2.83	2.99 ± 0.01
		H2H2 (64)	10,560 ^ABb^ ± 108.83	379.53 ^ABb^ ± 4.56	3.60 ± 0.04	319.76 ^AB^ ± 3.32	3.04 ± 0.02
		H2H3 (85)	10,971 ^Aa^ ± 98.06	385.25 ^ABab^ ± 4.15	3.52 ± 0.04	329.89 ^A^ ± 3.02	3.01 ± 0.014
		*p* value	<0.0001	<0.0001	0.3899	<0.0001	0.1286
BLOCK2	1	H1H1 (118)	10,243 ^B^ ± 66.47	349.59 ^b^ ± 2.33	3.43 ± 0.04	309.81 ^B^ ± 2.90	3.00 ± 0.01
		H1H3 (71)	10,679 ^A^ ± 80.86	359.13 ^a^ ± 2.84	3.39 ± 0.04	323.14 ^A^ ± 3.30	3.02 ± 0.02
		H2H2 (278)	10,336 ^B^ ± 52.06	353.69 ^ab^ ± 1.83	3.45 ± 0.03	312.8 ^B^ ± 2.45	3.02 ± 0.01
		*p* value	<0.0001	0.017	0.329	0.0004	0.3323
	2	H1H1 (79)	10,988 ^Aa^ ± 108.25	391.81 ^Aa^ ± 4.58	3.58 ± 0.04	327.91 ^a^ ± 3.33	2.99 ± 0.02
		H1H3 (50)	10,544 ^Bb^ ± 131.80	369.04 ^Bb^ ± 5.51	3.51 ± 0.05	316.52 ^b^ ± 4.02	3.00 ± 0.02
		H2H2 (181)	10,597 ^Bb^ ± 83.65	379.16 ^ABb^ ± 3.64	3.58 ± 0.03	319.04 ^b^ ± 2.65	3.02 ± 0.01
		*p* value	0.0007	0.0006	0.4171	0.0088	0.1168
BLOCK3	1	H1H2 (239)	10,194 ± 73.90	345.47 ^b^ ± 3.22	3.40 ± 0.03	305.64 ^B^ ± 2.34	3.00 ± 0.01
		H2H2 (580)	10,292 ± 62.97	346.25 ^b^ ± 2.82	3.38 ± 0.03	308.43 ^AB^ ± 2.05	3.00 ± 0.01
		H2H3 (48)	10,439 ± 118.76	357.32 ^a^ ± 4.93	3.43 ± 0.05	315.61 ^A^ ± 3.59	3.03 ± 0.02
		*p* value	0.06	0.0315	0.3858	0.0103	0.2938
	2	H1H2 (176)	10,456 ^Ab^ ± 79.36	312.66 ^b^ ± 3.40	3.58 ± 0.03	312.67 ^C^ ± 2.50	2.99 ± 0.01
		H2H2 (370)	10,700 ^Aa^ ± 67.79	320.34 ^a^ ± 2.97	3.55 ± 0.03	320.34 ^B^ ± 2.19	3.00 ± 0.01
		H2H3 (37)	10,860 ^Aa^ ± 138.53	327.69 ^a^ ± 5.71	3.60 ± 0.06	327.69 ^A^ ± 4.18	3.02 ± 0.02
		*p* value	0.0005	0.0052	0.3623	<0.0001	0.496

The number in the table represents the mean ± standard deviation; the number in the bracket represents the number of cows with the corresponding genotype; the *p* value shows the significance for the genetic effects of SNPs; the superscript letters indicate the significance calculated for the effects of different genotypes via the comparison of each pair of pairs; ^a^, ^b^ within the same column with different superscripts indicates *p* < 0.05; and ^A^, ^B^, ^C^ within the same column with different superscripts indicates *p* < 0.01.

**Table 5 animals-14-01276-t005:** Transcription factor binding sites (TFBSs) prediction for *CYP7A1* and *HADHB* genes.

Gene	SNP	Allele	Transcription Factor	Relative Score (≥0.80)	Predicted Binding Site Sequence
*CYP7A1*	14:g.24676921A>G	A	-	-	-
		G	EBF1	0.89	ATTCCAGGGA
			SP1	0.88	GGGACGGGG
	14:g.24675708G>T	G	ELK1	0.80	GGCACTGAAA
		T	FOXC1	0.82	TTTGTAAATGC
			ZNF354C	0.82	ATGCAC
*HADHB*	11:g.73256269T>C	T	PAX2	0.84	GGTCGTGC
			NR2F1	0.82	GAAGGGTCGTGCG
		C	TFAP2A	0.92	GCCGTGCGC

Relative scores represent the correlation score for each predicted binding site, with higher values indicating a stronger possibility of binding. The SNP in the predicted binding-site sequence is underlined.

## Data Availability

The datasets generated and/or analyzed during the current study are available in the article and its Appendix A.

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
