# Peer review of "Polymorphisms of CYP7A1 and HADHB Genes and Their Effects on Milk Production Traits in Chinese Holstein Cows"

_animals, 2024, doi:10.3390/ani14091276_

Round 1
Reviewer 1 Report
Comments and Suggestions for Authors
1、Preliminary research proposed CYP7A1 and HADHB genes as candidates for milk production traits in dairy cattle.Based on this, further identification of SNPs in these two genes was conducted to explore the association between the SNPs within the genes and phenotypes, as well as their effects on the genes. The research idea is clear, the experimental design is reasonable, and the results are clearly presented.
2、However, the writing details need further checking and improvement, such as the presence of an underline under the word "of" in line 119.
3、In section 3.1, the identification of SNPs is based on the results of gene amplification using how many individuals as templates, and whether these individuals are sufficiently representative should be clarified.
4、It is suggested to remove the mention in the conclusion that SNPs affect mRNA stability. This part is only a prediction result and is not recommended to be included in the conclusion.
Author Response
Re: Animals-2978374- Comments from Reviewers
Dear Reviewers,
Thank you very much for your time involved in reviewing the manuscript and your very encouraging comments on the merits.
Comments:
“Preliminary research proposed CYP7A1 and HADHB genes as candidates for milk production traits in dairy cattle. Based on this, further identification of SNPs in these two genes was conducted to explore the association between the SNPs within the genes and phenotypes, as well as their effects on the genes. The research idea is clear, the experimental design is reasonable, and the results are clearly presented.”
We also appreciate your clear and detailed feedback and hope that the explanation has fully addressed all of your concerns. In the remainder of this letter, we discuss each of your comments individually along with our corresponding responses.
To facilitate this discussion, we first retype your comments in italic font and then present our responses to the comments.
Point 1: However, the writing details need further checking and improvement, such as the presence of an underline under the word "of" in line 119.
Response 1:
Thanks for your comments and suggestions. As suggested, we have removed the underline under the word "of" in line 119 and checked the writing details in the whole manuscript (please see page 3, line 119; page 2, lines 77; page 11, lines 255-257; page 3, lines 113 and page 4, lines 167).
Point 2: In section 3.1, the identification of SNPs is based on the results of gene amplification using how many individuals as templates, and whether these individuals are sufficiently representative should be clarified.
Response 2: Thanks for your comments. Done as suggested, we have added the descriptions of individuals and their sufficiently representative (please see page 4, lines 172-173).
Point 3: It is suggested to remove the mention in the conclusion that SNPs affect mRNA stability. This part is only a prediction result and is not recommended to be included in the conclusion.
Response 3: Done as suggested (please see page 9, line 310-311).
Finally, we wish to thank you again for the critical comments and valuable suggestions. If you have any question on this manuscript, please feel free to contact me.
Thank you again for your time and favorable consideration.
Sincerely
Bo Han, Ph.D.
Tel: +86 13810209681
Email: [email protected]

Reviewer 2 Report
Comments and Suggestions for Authors
The manuscript shows the effects of CYP7A1 and HADHB polymorphisms on milk, fat, and protein yield in Chinese Holstein cows. Several SNPs and their haplotype blocks were identified in both genes. Individual SNP and haplotype effects were detected on milk, fat and protein production. Also, biochemical action of these SNPs has been assessed.
It is noteworthy that this manuscript is based on previous research where both genes showed differential expression between different stages of lactation in liver tissues of Chinese Holstein cows and potential roles in lipid metabolism. Therefore, it constitutes a very interesting and well-based continuation of the previous studies. On the other hand, this study would suppose a good basis to extend existing Genomic Selection arrays for dairy cattle breeding.
Objectives were clear and well described. Methods have been described in detail and were adequate to objectives. Conclusions are very well based on results. Perhaps mention should be made of potential subsequent verification in another study group.
References list includes a total of 29 references (12/29=0.41 from the last 10 years, 3/29=0.10 from the last 5 years).
Specific comments:
Tables 1 and 2 are difficult to read; attention must be paid to alignment of lines. This problem is less pronounced in Table 3, perhaps due to the presence of several horizontal lines. On the other hand, in Table 2, some superscripts are missing in significant comparisons. Supplementary tables contain interesting information and are useful for better understanding of this study.
Figure 1 also shows a complete linkage disequilibrium for 14: g24676921A>G with both 14:g.24664026A<G and 14:g.24665961C>T. However, no reference is made to this situation nor any explanation is provided.
Author Response
Re: Animals-2978374- Comments from Reviewers
Dear Reviewers,
Thank you very much for your time involved in reviewing the manuscript and your very encouraging comments on the merits.
Comments:
“The manuscript shows the effects of CYP7A1 and HADHB polymorphisms on milk, fat, and protein yield in Chinese Holstein cows. Several SNPs and their haplotype blocks were identified in both genes. Individual SNP and haplotype effects were detected on milk, fat and protein production. Also, biochemical action of these SNPs has been assessed. It is noteworthy that this manuscript is based on previous research where both genes showed differential expression between different stages of lactation in liver tissues of Chinese Holstein cows and potential roles in lipid metabolism. Therefore, it constitutes a very interesting and well-based continuation of the previous studies. On the other hand, this study would suppose a good basis to extend existing Genomic Selection arrays for dairy cattle breeding. Objectives were clear and well described. Methods have been described in detail and were adequate to objectives. Conclusions are very well based on results. Perhaps mention should be made of potential subsequent verification in another study group.”
We also appreciate your clear and detailed feedback and hope that the explanation has fully addressed all of your concerns. In the remainder of this letter, we discuss each of your comments individually along with our corresponding responses.
To facilitate this discussion, we first retype your comments in italic font and then present our responses to the comments.
Point 1: Tables 1 and 2 are difficult to read; attention must be paid to alignment of lines. This problem is less pronounced in Table 3, perhaps due to the presence of several horizontal lines. On the other hand, in Table 2, some superscripts are missing in significant comparisons. Supplementary tables contain interesting information and are useful for better understanding of this study.
Response 1:
Thanks for your comments and suggestions. As suggested, we have modified Table 1 and 2, and the reasons of some superscripts are missing in variance test there were some significant differences overall, but no significant differences in two-by-two comparisons. This may be explained by the more conservative correction for p-value in two-by-two comparisons using Bonferroni's method to correct for p-value. (please see page 4 and 5, line 182-183; page 3, lines 197-204). And we have placed table S3 into the manuscript (please see pages 9 and 10, line 204-205).
Point 2: Figure 1 also shows a complete linkage disequilibrium for 14: g24676921A>G with both 14:g.24664026A<G and 14:g.24665961C>T. However, no reference is made to this situation nor any explanation is provided.
Response 2: Thanks for your comments. We found that between 14: g24676921A>G and 14:g.24664026A<G, although showing a higher D' = 1, its r2 is only 0.28 (please see figure below), which is difficult to explain that the two SNPs are indeed in linkage disequilibrium. So we didn't put 14: g24676921A>G in block 1. The picture in the doc may help me explain this data better.
Finally, we wish to thank you again for the critical comments and valuable suggestions. If you have any question on this manuscript, please feel free to contact me.
Thank you again for your time and favorable consideration.
Sincerely
Bo Han, Ph.D.
Tel: +86 13810209681
Email: [email protected]
